# Damage Detection Using Modal Rotational Mode Shapes Obtained with a Uniform Rate CSLDV Measurement

**Zi Huang** and **Chaoping Zang** *

Jiangsu Province Key Laboratory of Aerospace Power System, College of Energy and Power Engineering, Nanjing University of Aeronautics and Astronautics, Nanjing 210016, China; huangzi_chris@163.com
* Correspondence: c.zang@nuaa.edu.cn; Tel.: +86-025-8489-2200

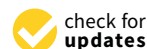

**Featured Application: This approach has a potential application of non-contact structural health monitoring on ultra-light structures or plate-like structures with composite materials based on measured vibration data.**

**Abstract:** With the rapid development of a continuously scanning laser Doppler vibrometer (CSLDV) technique, the full-field mode shapes of structures with high accuracy can be obtained. In this paper, a novel damage detection method using modal rotational mode shapes obtained with a uniform rate CSLDV measurement is proposed. The modal rotational damage indicators considering the changes of modal rotational mode shapes between the damaged and the undamaged states are established. Because the modal rotational mode shapes are obtained through the derivative of the detailed displacement mode shapes of transitional degree-of-freedoms (DOFs) with respect to the orthogonal directions, they are more sensitive than the normal displacement mode shapes. The uniform rate CSLDV measurement is essentially a uniform straight-line scanning technique and the measured mode shapes can be directly obtained through the demodulation of vibration signals. Besides, taking it for granted that a priori knowledge of the undamaged structure is not known, the undamaged mode shapes can be reconstructed from the measured damaged data using the fitted polynomial functions in which the minimum number of polynomial function coefficients are determined by a fit value threshold. The proposed method is firstly demonstrated by numerical simulation of the crack plate and then a plate structure with three damaged cases is taken as an example for further experimental study. The experimental results indicate the following: (1) The uniform rate CSLDV measurement can obtain the high accuracy modal rotational mode shapes with the advantage of eliminating the contaminated noise in the measurement; (2) the modal rotational damage indicators of the torsional modes are the most sensitive to the crack damage and they can clearly identify single, multiple damages and locations of the plate, and even slight crack damage, respectively. The effectiveness of the method paves the way for practical applications, such as ultra-light or composite structures.

**Keywords:** damage detection; modal rotational mode shape; modal rotational damage indicator; a uniform rate CSLDV; the fitted polynomial functions; ODS

## 1. Introduction

Damage detection of structures from changes in their vibration characteristics is widely used in engineering. The basic approach is to compare the modal characters between the damage and the undamaged states of the object structure. In the last three decades, many methods have been developed. The earliest damage detection methods are based on mode frequency changes of structures [1–3].

However, the mode frequency changes usually reflect the global features and are easily buried by the operational conditions. Afterwards, the detection methods using mode shape changes [1,2] are proposed to identify the crack or damage of structures. Later on, the change of modal flexibility of a few lower orders of mode shapes [4–6] were used for damage detection of structures in civil engineering. However, the displacement mode shape can be easily affected by environmental noise and is not sensitive to slight damage. Due to these reasons, methods based on the curvature mode shape [7–9] or mode strain energy [10–12] have been further developed and have become the most widely used in practice. This is because the damage indicator based on curvature mode shape or mode strain energy is much more sensitive and can easily localize anomalous features of the structure.

In general, the modal strain energy and mode curvature are often applied in combination to damage detection. Hu et al. [13–15] used the differential quadrature method (DQM) to calculate the modal strain energy to detect the crack of a plate or a circular hollow cylinder structure. Fan et al. [16] proposed a damage severity correction factor (DSCF) based on modal strain energy to reflect the nonlinear relationship between the indicator value and severity of the damage. Also, mode curvatures were measured using a surface-bonded polyvinylidene fluoride (PVDF) piezoelectric strain sensor array to detect the crack of a fiber-reinforced plastic (FRP) sandwich deck panel. However, the quality of mode curvature or modal strain energy achieved in practice depends on the spatial density and the accuracy of displacement mode shapes. Sazonov et al. [17] proposed an optimal spatial sampling interval method to minimize the effects of noise and truncation errors on the calculation of the curvature mode shape or modal strain energy. Cao et al. [18] developed a novel Laplacian operator and smoothed Teager energy operator (STEO) to eliminate the influence of data noise on mode curvature. Also, Cao et al. [19] formulated a new concept of complex-wavelet mode curvature to reveal and delineate the damage under noisy condition. The mode curvature can be expressed as the second order partial derivatives of the displacement mode shape. The second order differential will greatly amplify the noise effect. Not only that, the spatial resolution of the mode shape also has a great influence [17,20]. Due to these reasons, measurement noise needs to be strictly controlled and spatial resolution of displacement mode shape should be carefully considered to adapt to various damage detection with these methods.

In recent years, the full field vibration measurement techniques, such as continuously scanning laser Doppler vibrometer (CSLDV) [21,22] and high speed three-dimensional digital image correlation (DIC) [23–25] techniques, have become mature and widely used. Compared with traditional transducers, the full field techniques can accurately realize noncontact surface measurement and the spatial resolution of mode shape can be greatly improved. For the CSLDV method, there are two ways to scan the structural surface: (a) A uniform rate scan [26], and (b) a sinusoidal rate scan [27]. The demodulation of CSLDV output signals to obtain the operational deflection shape (ODS) is also different according to different scan strategies. For a sinusoidal rate scan method, the displacement mode shape is fitted using lower order polynomial function based on the sideband information of CSLDV output signals. Hence, the slight changes of mode shapes caused by damage are easily lost during the fitting process, especially for the lower modes. On the contrary, using a uniform rate scan method, the displacement mode shapes can be directly obtained by the demodulation of CSLDV output signals at the corresponding spatial position. Based on that, Chen et al. [28,29] and Hu et al. [30] used the uniform rate scan CSLDV method to measure the ODS and detect the crack of beam or delamination of composite plate. For the DIC method, mode curvature using measured ODS is derived to detect the crack of sandwich panel and membranes [31,32].

Actually, advantages of damage detection using the mode curvature and modal strain energy methods are due to the fact that they are based on the second order differential of the displacement mode shape which can significantly amplify the slight discontinuous change caused by the damage. In the same way, the modal rotation, which is expressed as the first order partial derivative of displacement, has the similarity features with mode curvature and therefore can also be applied to damage detection. Katunin et al. [33] used the measured modal rotation with DIC method to detect the crack of a beam

with the combination of wavelet analysis. Xu et al. [34] proposed a damage detection method for beam structures using the slopes of displacement mode shapes from CLSDV. In addition, Kim et al. [35] proposed a strain based rotational mode shape measurement method for beam structures. The rotational shape can be derived by integrating the strain mode shape. Based on that, the long-gauge strain measurement technique is developed for structural damage detection and health monitoring, especially for long span bridges [36,37]. The modal rotational mode shape is sensitive to reflect the discontinuous change caused by the damage and is also less sensitive to the noise contamination in the mode shape. Therefore, it is a good balanced parameter suitable for structural damage detection.

Another issue for damage detection is that mode shapes of the undamaged state of the structure are hardly obtained in many cases. This causes difficulty in comparing the mode shape changes between the damaged and the undamaged states. Hoerst and Ratcliffe [38] developed a finite difference approximation of Laplace's differential operator to the mode shape from the damaged structure in order to identify and pinpoint structural damage in a beam without a priori knowledge of the undamaged structure. This procedure operates solely on mode shape data of damaged state. Later on, Ratcliffe [39] used a modified Laplacian operator on mode shape data to locate and estimate the crack of Euler-Bernoulli beam structures. Yoon et al. [40] developed a two-dimensional gapped smoothing method (GSM) to detect the plate damage based on Ratcliffe's method. Limongelli [41] proposed a two-dimensional surface interpolation method to localize reductions of stiffness in plate-like structures. Most recently, Chen et al. [28,42] proposed a polynomial function fitted method to the mode shape of the damaged state in order to proximately estimate the mode shape of the undamaged structure. Zang [43] and Cao et al., [44] used the measured mode shape data of the damaged state with principal component analysis (PCA) method to reconstruct the mode shapes of the undamaged structure. The polynomial function fitted ODSs or mode shapes are relatively smooth and can reflect the property of the undamaged structure.

In this paper, a damage indicator based on changes of the modal rotational mode shapes between the estimated undamaged and the tested damaged states using a uniform rate CSLDV measurement of the only damaged plate-like structure is put forward. The modal rotational mode shapes are obtained through the derivative of the detailed displacement mode shape of transitional degree-of-freedoms (DOFs) with respect to the orthogonal directions. The undamaged mode shapes were reconstructed with fitted polynomial functions in which the minimum number of polynomial function coefficients are determined by a fit value threshold. Taking a plate structure with various damaged cases as an example, the ODS data of the damaged plate at resonance frequencies of the first several modes are obtained using the CSLDV system and the corresponding ODSs of the undamaged state are also reconstructed. After that, modal rotational mode shapes and rotational damage indicators are derived and plotted to identify the damage locations of the plate structure.

The rest of the paper is organized as follows. The theory of the scan strategy and mode shape demodulation method, modal rotational damage indicators are introduced in Section 2. Then, numerical investigation of a plate structure is presented and analyzed in Section 3. In Section 4, the effectiveness of the proposed damage detection method is experimentally demonstrated through three different cases and the results are discussed. Finally, conclusions are drawn in Section 5.

## 2. Methodology of Damage Detection Using Continuously Scanning Laser Doppler Vibrometer (CSLDV) Measurement Data

### 2.1. Methodology of Operational Deflection Shape (ODS) and Mode Shape

The basic equation of motion for a vibration structure which was excited by a force vector can be expressed as

$$M\boldsymbol{x}''(t) + C\boldsymbol{x}'(t) + K\boldsymbol{x}(t) \ = \ \boldsymbol{f}(t) \tag{1}$$

where vector $x(t)$ consists of the spatial coordinates describing the motion of the structure. The $M, K, C$ represent the global mass, stiffness, and damping matrices of the structure, respectively. The vector $f(t)$ is the excitation force. For the harmonic excitation force, the force function can be written as

$$f(t) = Fe^{i\omega t} \tag{2}$$

Thus, the corresponding harmonic response of the system is expressed as

$$x(t) = Xe^{i\omega t} \tag{3}$$

Substituting Equations (2) and (3) into Equation (1), the force response solution can be directly written as

$$X = (K - \omega M + i\omega C)^{-1}F = H(\omega)F \tag{4}$$

The frequency response matrix $H(\omega)$ can be derived from the modal parameters and a more explicit form of the force response solution may be expressed as follows:

$$X = \sum_{r=1}^{N} \frac{\varphi_r^T F \varphi_r}{(\omega_r^2 - \omega^2 + i2\xi_r\omega_r\omega)} \tag{5}$$

where $\varphi_r, \omega_r, \xi_r$ represent the mode shape, natural frequency, and modal damping ratio of the $r^{th}$ mode, respectively. Equation (5) permits the calculation of one or more individual responses to an excitation of several simultaneous harmonic forces. The modal response $X$ is sometimes referred to as an operating deflection shape (ODS). It can be seen that the ODS is a function of both modal properties and the excitation forces and can be written as the combination of real and imaginary parts:

$$X = X_R + iX_I \tag{6}$$

where $X_R, X_I$ are the real and imaginary parts of ODS. When a sinusoidal excitation frequency is equal to the $r^{th}$ modal frequency of the system, the $r^{th}$ mode shape is almost dominated the ODS. Therefore, in this case, the real part of ODS $X_R$ can be approximately treated as the $r^{th}$ mode shape.

*2.2. Continuously Scanning Strategy for Damage Detection*

The CSLDV system contains a single head scanning vibrometer and an extra National Instruments (NI) and PCI extensions for Instrumentation (PXI) system for providing control signals to manipulate the two orthogonal scanning mirrors to cover the scanning rectangle area. Generally, there are two continuously scanning strategies: The Lissajous and the Zigzag, as shown in Figure 1. For Lissajous scanning, the sinusoidal signals with two different frequencies are used to drive the orthogonal scan mirrors. The main purpose of demodulation is to obtain the sideband information and fit the mode shape of the plate using polynomial functions based on the sideband spectral data. However, the changes of mode shape caused by the damage or crack are usually local and could be easily treated as noise and eliminated through the mode shape fitting process. On the contrary, for the Zigzag scanning, the control signals to the scanning mirrors are triangular waves or saw tooth signals in order to make the laser spot scan along a line path and span a rectangle scan area. The Zigzag scanning is essentially a uniform straight-line scan method. The measured mode shape can be directly obtained through the demodulation of vibration signals with spatial position transformed to the time axis. Then, the mode shape can be realized by plotting this demodulated signal as a three-dimensional surface against the measurement position, which is derived from the simultaneously recorded mirror drive signals.

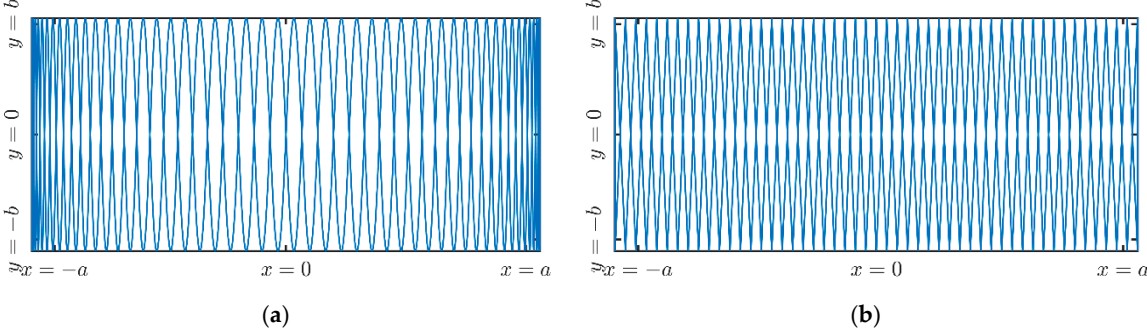

**Figure 1.** Two different scan strategies of area scanning. (**a**) Lissajous strategy with a sinusoidal rate scan. (**b**) Zigzag strategy with a uniform rate scan.

Here, the uniform rate continuously scanning method is introduced to perform a rectangle area measurement. The boundaries of the scan area are $x \in [x_{\min}, x_{\max}]$, $y \in [y_{\min}, y_{\max}]$. The spatial resolution of measured mode shape along the scanning direction is determined by the sample frequency and the scan frequency. In the x direction, the sample frequency $f_s$ and the scan frequency $f_x$ can be expressed as

$$SR_x = 2(x_{\max} - x_{\min})f_x/f_s \tag{7}$$

In the y direction, it can be treated as stepped scan and the real spatial resolution can be expressed as

$$SR_y = (y_{\max} - y_{\min})/(f_x/(2f_y)) \tag{8}$$

In order to cover the measured plate as much as possible, the scan parameters should be reasonably adjusted according to geometrical characteristics of the test structure and requirement of the measurement. In general, lower scan frequency and higher sample frequency will lead to high resolution of the mode shape data. But it will take a longer scanning time and will generate a large amount of vibration signal data, which makes data demodulation more difficult.

### 2.3. Mode Shapes by Demodulation of Uniform—Rate Scanning CSLDV Output Signals

#### 2.3.1. Measured Mode Shapes of the Damaged State

For a linear structure subjected to sinusoidal excitation at the $r^{th}$ modal frequency $\omega_r$, the measured velocity vibration $V(x, y, t)$ of a specific point $P(x, y)$ perpendicular to the surface can be expressed as

$$V(x, y, t) = V_R(x, y)\cos(\omega_r t) + V_I(x, y)\sin(\omega_r t) \tag{9}$$

A convenient method for demodulation is simply to multiply the CSLDV output signals by in-phase and quadrature signals at the excitation frequency, respectively, and given as

$$
\begin{aligned}
V(t)\cos(\omega_r t) &= V_R(t)\cos^2(\omega_r t) + V_I(t)\sin(\omega_r t)\cos(\omega_r t) \\
&= \tfrac{1}{2}V_R(t) + \tfrac{1}{2}V_R(t)\cos(2\omega_r t) + \tfrac{1}{2}V_I\sin(2\omega_r t) \\
&= \ldots(LPF)\ldots = \tfrac{1}{2}V_R(t)
\end{aligned} \tag{10}
$$

$$
\begin{aligned}
V(t)\sin(\omega_r t) &= V_R(t)\cos(\omega_r t)\sin(\omega_r t) + V_I(t)\sin^2(\omega_r t) \\
&= \tfrac{1}{2}V_I(t) + \tfrac{1}{2}V_R(t)\sin(2\omega_r t) - \tfrac{1}{2}V_I(t)\cos(2\omega_r t) \\
&= \ldots(LPF)\ldots = \tfrac{1}{2}V_I(t)
\end{aligned} \tag{11}
$$

where the LPF represents the low-pass filter. Because the scan frequency is slow enough, the signal components at frequency $2\omega_r$ are removed with the LPF and the real and imaginary parts of ODS can be obtained. Practically, due to the mechanical delay of the scanning mirror, the actual measurement position of the laser spot deviates from the ideal position given by the control signal. Before the mode

shape demodulation, the position signal needs to be offset to match the vibration signal as much as possible. Moreover, we need to adjust the signal phase so that the real and imaginary parts of ODS attain their maximum and minimum amplitudes. It can be expressed as

$$V_R(x,y) = V(x,y)\cos(\alpha), V_I(x,y) = V(x,y)\sin(\alpha) \tag{12}$$

where $\alpha$ is the optimized phase value. In this case, the corresponding real part of ODS $V_R$ is the mode shape.

### 2.3.2. Surface Fitting of the Undamaged Mode Shapes

In practical applications, mode shapes of the undamaged structure are usually non-existent or difficult to obtain. Most of the damage only causes local changes of the mode shape—the overall effect on the mode shape is not obvious. Based on the above reasonable assumption, we can use the binary polynomial function to fit the overall mode shape using measured shape data, and it can be regarded as the mode shape of undamaged state. Assuming the measured mode shape of damaged state can be expressed by binary polynomial functions approximately:

$$V_R(x,y) = \sum_{m=0,n=0}^{p.q} W_{(m,n)}\overline{x}^m\overline{y}^n \tag{13}$$

where, $p, q$ are the determined polynomial function orders, $\overline{x}, \overline{y}$ are the central normalized coordinates of the measured plate to avoid the ill condition problem, $W_{(m,n)}$ is the corresponding polynomial coefficient. The polynomial coefficients can be solved using the least-square method

$$W_{(p+1)\times(q+1),1} = \left(U_{N,(p+1)\times(q+1)}\right)^{-1}V_R \tag{14}$$

where, $U$ is the Vandermonde matrix of the central normalized coordinates, $N$ is the point number of the measured mode shape. The Vandermonde matrix can be expressed as

$$U = \begin{bmatrix} 1 & \cdots & \overline{x}_1^p\overline{y}_1^q \\ \vdots & \ddots & \vdots \\ 1 & \cdots & \overline{x}_N^p\overline{y}_N^q \end{bmatrix} \tag{15}$$

However, the number of measured points using CSLDV are huge. In most cases, it is larger than hundreds of thousands or more. In order to avoid too much computation and over fitting, a proper order of the polynomial function needs to be determined. In order to evaluate the similarity between the fitting result and the original result, a fit value is defined as

$$fit(p,q) = \frac{rms(V_R)}{rms(V_R) + rms(e_{p,q})} \times 100\% \tag{16}$$

where $e = U_{N,p\times q}W_{p\times q,1} - V_R$ is the deviation between the measured mode shape and fitted mode shape using the polynomial function, and $rms()$ means the root-mean-square value. In order to determine the polynomial order, a convergence index is defined as

$$\begin{cases} con(p) = fit(p,q) - fit(p-2,q) \\ con(q) = fit(p,q) - fit(p,q-2) \end{cases} \tag{17}$$

When the convergence index is less than the threshold, the polynomial order is determined. In this paper, the threshold value is set to be 0.1% to keep the best fitting effect.

### 2.4. Modal Rotational Damage Indicators

The modal rotational mode shape of the measured plate can be estimated using the normal translational mode shape. The modal rotation of point $(x, y)$ can be expressed by the central finite-difference method as below:

$$
\begin{aligned}
\theta_x(x, y) &= \frac{V_R(x, y + \delta y) - V_R(x, y - \delta y)}{2\delta y} \\
\theta_y(x, y) &= -\frac{V_R(x + \delta x, y) - V_R(x - \delta x, y)}{2\delta x}
\end{aligned}
\tag{18}
$$

where $\delta x, \delta y$ are the distances between two neighboring measurement points along the x and y directions, respectively. When the point locates near the boundary of the scan area or the edge of plate, the forward or backward finite-difference formulation can be used to estimate corresponding modal rotation and expressed as

$$
\begin{aligned}
\theta_x(x, y) &= \frac{3V_R(x, y - \delta y) - 4V_R(x, y) + V_R(x, y + \delta y)}{2\delta y} \\
\theta_y(x, y) &= -\frac{3V_R(x - \delta x, y) - 4V_R(x, y) + V_R(x + \delta x, y)}{2\delta x}
\end{aligned}
\tag{19}
$$

The modal rotational mode shape of the undamaged state also can be derived using the above formula. Then, the modal rotational damage indicator (RDI) is defined as the changes of modal rotation in both directions, that is

$$
RDI_x = \left| \theta_x^U - \theta_x^D \right|, \quad RDI_y = \left| \theta_y^U - \theta_y^D \right|
\tag{20}
$$

The superscripts $U, D$ represent the undamaged and the damaged states, respectively.

## 3. Numerical Investigation

### 3.1. Finite Element Analysis

The finite element model of a plate is used to perform the demonstration of damage detection with the modal rotational mode shapes. The length of the plate along the *x*-axis is 180.0 mm and width along the *y*-axis is 80.0 mm. The thickness of the plate is 2.0 mm. Material property of the plate is assumed to be the same as the steel with Young's modulus $E = 200\text{GPa}$, Poisson's rate $v = 0.3$, and density $\rho = 7800\text{Kg}/\text{m}^3$. The plate has clamped-free-free-free boundary conditions and is fixed at the left edge. It is divided into $180 \times 80$ quadrilateral shell elements of size 1 mm. There are two different length cracks located at the plate and span along the y direction. The first crack span length is 15 mm, approximately equal to 18.75% of the width of the plate and the second is 25 mm, which equals 31.25% of the width, as shown in Figure 2. Both cracks are called Crack 1 and Crack 2 hereafter, respectively. In order to demonstrate the impact of damage, the thickness at the damage area is reduced to 0.1 mm.

Normal modal analysis of the damaged plate is performed and the first four displacement mode shapes are extracted and shown in Figure 3. The first and third modes are pure bending modes, while the second and fourth modes are torsional modes. For the first mode, the discontinuous of the displacement mode shape caused by the damage is not obvious and hardly reflects the location of the crack. Only Crack 2 can be reflected through the third mode shape and is shown in Figure 3c. On the country to the bending modes, the discontinuous of mode shapes of torsional modes at the crack location are more significant, as the second and fourth modes are shown in Figure 3b,d.

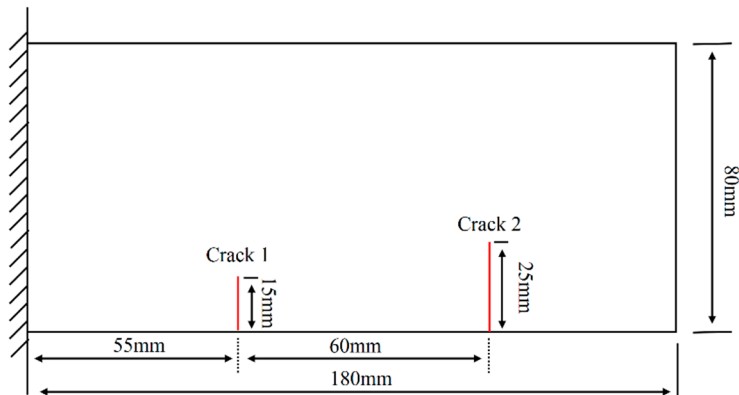

**Figure 2.** The geometry of a damaged plate structure.

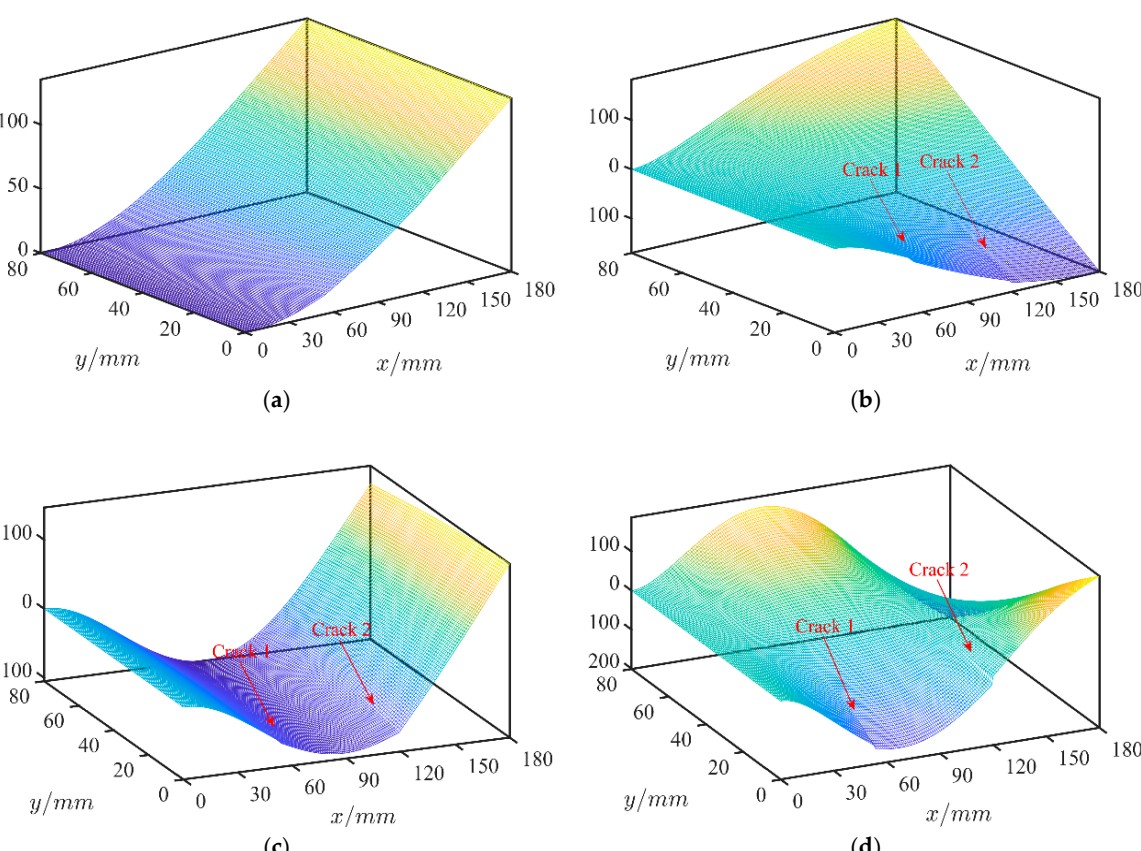

**Figure 3.** The first four mode shapes of normal translation degree-of-freedoms (DOFs) of the damaged plate model. (**a**) The first mode. (**b**) The second mode. (**c**) The third mode. (**d**) The fourth mode.

### 3.2. Damage Detection with Modal Rotational Damage Indicators

In order to have a comparison between the displacement mode shape and the rotational mode shape for damage detection, the changes of displacement mode shapes between the damaged and the undamaged states are firstly used to identify the crack. The color plots of these changes of the first four mode shapes are shown in Figure 4. For the first mode, the identified results contain not only the approximate locations of the two cracks but also the free edge area of the right side of the plate. The change of the second mode shape indicates the large damage areas between the two cracks along the x direction and the free edge area of the right side of the plate although the heavier damage location is obvious. For the third and fourth modes, both changes can indicate multiple possible damage areas including the real damage places, but still fail to flag out the exact damage locations. Figure 4 also indicates that the displacement mode shapes of lower modes are not sensitive to the damage locations.

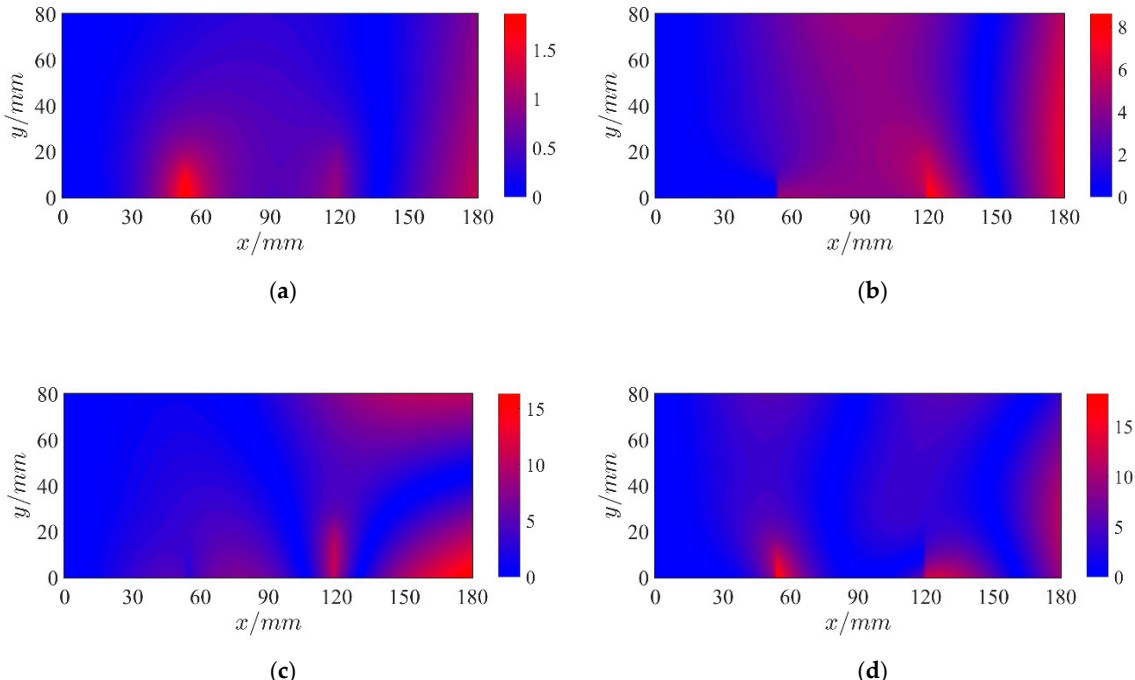

**Figure 4.** The colored plot of changes of the first four displacement mode shapes. (**a**) The first mode. (**b**) The second mode. (**c**) The third mode. (**d**) The fourth mode.

The color plot of the first four modal rotational damage indicators of both the x and y directions are listed in Figure 5, with the normalization by the maximum value of the damage indicators of both directions. It clearly shows that the modal rotational indicators of both directions can correctly flag out the damaged areas of the plate. The modal rotational indicators of the y direction are significantly larger than the results of the x direction with the detected damage areas more accurate. The reason can be attributed to the fact that the crack span direction is along the y axis, and the crack will cause vibration separation perpendicular to the crack span direction and the modal rotation will be increased significantly at the crack location. Therefore, the second and fourth torsional modes are the most sensitive to the damage locations. Their modal rotational indicators of the y direction exactly localized the crack positions and the span directions.

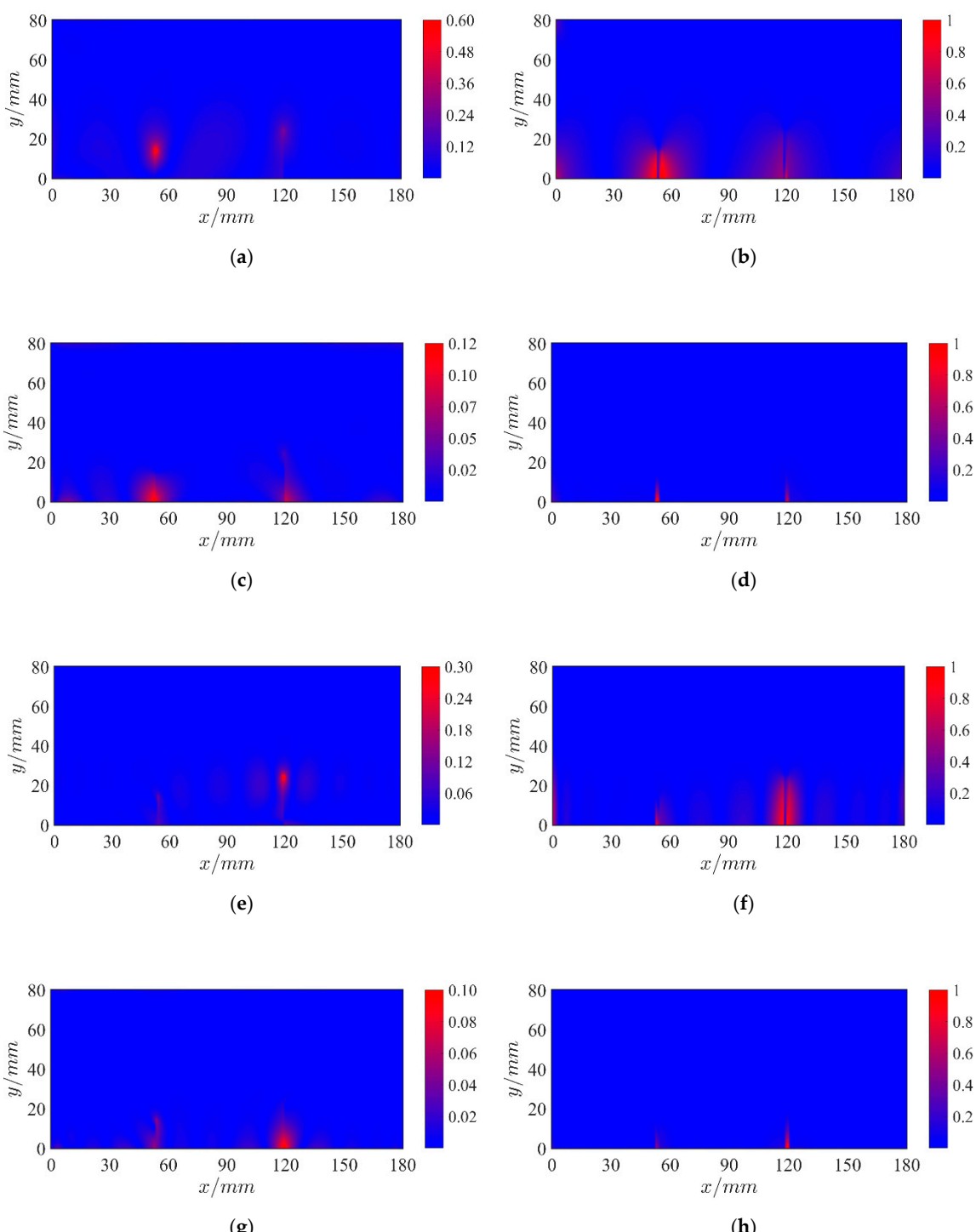

**Figure 5.** The rotational damage indicators (RDIs) colored plot of the first four modes. (**a**) RDIs of the first mode of the x direction. (**b**) RDIs of the first mode of the y direction. (**c**) RDIs of the second mode of the x direction. (**d**) RDIs of the second mode of the y direction. (**e**) RDIs of the third mode of the x direction. (**f**) RDIs of the third mode of the y direction. (**g**) RDIs of the fourth mode of the x direction. (**h**) RDIs of the fourth mode of the y direction.

## 4. Experimental Validation

### 4.1. Experimental Configuration

The test piece in this paper was a steel plate with the same size and material properties used for numerical analysis. In order to reduce the measurement noise, a reflective sheeting was pasted on the measured surface of the plate to maximize the backscatter of laser light. The plate was clamped at the bottom edge with four M8 bolts and excited via a sound speaker near the top edge. The sound speaker is located close to the plate in order to ensure the plate can be excited. The CSLDV system consists of a Polytec PSV-400 LDV scan head, NI PXI system and an amplifier. The scanning mirrors in the Polytec LDV scan head is controlled by an analogue voltage input and output card in the additional NI PXI system in order to manipulate the continuous measurement of the rectangle area. The vibration signals are also captured by the NI PXI system. A sinusoidal excitation signal was generated by the junction box of Polytec system and was amplified by the amplifier before sending to the sound speaker. The control of all the hardware and signal processing were accomplished by the LabVIEW codes. The test configuration, scan head of LDV and NI PXI system are plotted in Figure 6a.

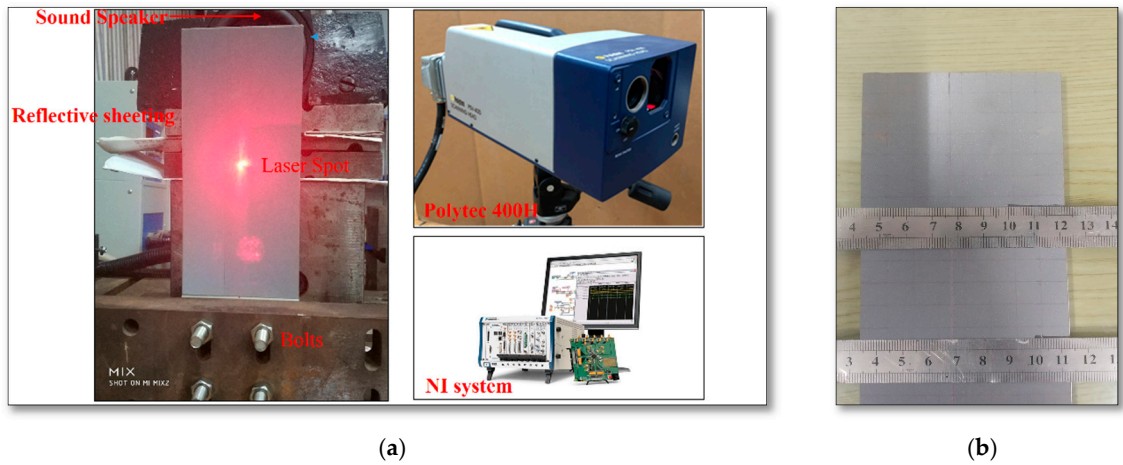

(**a**)            (**b**)

**Figure 6.** The test configuration. (**a**) The test configuration, scan head of laser doppler vibrometer (LDV) and National Instruments (NI) and PCI extensions for Instrumentation (PXI) system. (**b**) Test piece of multi cracks.

In order to facilitate control of the scan path of scanning mirrors, the original position of the laser spot should be located at the center of the plate. Moreover, the distance between the scan head and the laser spot was measured and prepared for calculating rotation angles of the mirrors in order to cover the area of the scanned plate as much as possible. To avoid the control mirrors rotating too much, the distance should be larger than 1.5 m and the corresponding maximum rotated angle is less than 3.5 degrees for the test piece. In order to increase the spatial resolution, the sampling frequency rate is set to be 16,384 Hz. Scan rates of 0.05 Hz along the x-direction and 10.0 Hz along the y-direction were chosen to give a Zigzag trajectory with a good coverage of the objective surface. The scanning boundaries are (−88, 88) along the x direction and (−39, 39) along the y direction to avoid the laser spot moving out of the plate. With these parameters, the spatial resolutions along the x and y directions are about 1.7 mm and 0.05 mm, respectively. It only takes 20 s to complete the continuously scanning measurement.

Three damaged cases, described in Table 1, are considered and discussed to demonstrate the effectiveness of the proposed method. The first case is a single crack at Crack 1 position, similar with the simulation in Section 3.1. The crack length is 13 mm and approximately 16.25% of the width of the plate. The second is a multi-crack case with two different length cracks at different locations, as shown in Figure 6b. The first crack is the same as the first case and a second crack with 22 mm length is added

at Crack 2 position, which is more than 27.5% of the width. The last case is a slight crack with only 6 mm length (7.5% of the width) at the position of Case 1 to illustrate the ability to detect a slight damage.

**Table 1.** The detailed description of three damage cases.

| Damage Case | Location | Size | Percent Length |
| --- | --- | --- | --- |
| Case 1 | Crack 1 | 13 mm × 1 mm | 16.25% |
| Case 2 | Crack 1 & Crack 2 | 13 mm × 1 mm & 22 mm × 1 mm | 16.25% & 27.5% |
| Case 3 | Crack 1 | 6 mm × 1 mm | 7.5% |

At the beginning of the test, frequency response functions of several points were measured, and the first four mode frequencies of these damaged plates were obtained. These mode frequencies later were set as the excitation frequencies in uniform–rate scanning CSLDV tests to measure the corresponding mode shapes.

*4.2. Results and Discussions of Three Damaged Cases*

4.2.1. Case 1: A Single Crack

For Case 1, the first four mode frequencies were 49.02 Hz, 237.70 Hz, 311.52 Hz, and 781.25 Hz, respectively. For each mode, a sinusoidal signal of the mode frequency was sent to the speaker to generate the acoustic excitation. The uniform–rate scanning CSLDV tests were assigned on the damaged plate to measure the corresponding mode shapes. The displacement mode shapes at these frequencies are shown in Figure 7, respectively. Obviously, the mode shapes look dense and smooth due to the high resolution and anti-noise ability of the uniform–rate scanning CSLDV. However, the discontinuous change of the displacement mode shape can hardly be seen for the bending modes (modes 1 and 3), while the torsional modes show a clear abrupt change at the crack location (modes 2 and 4).

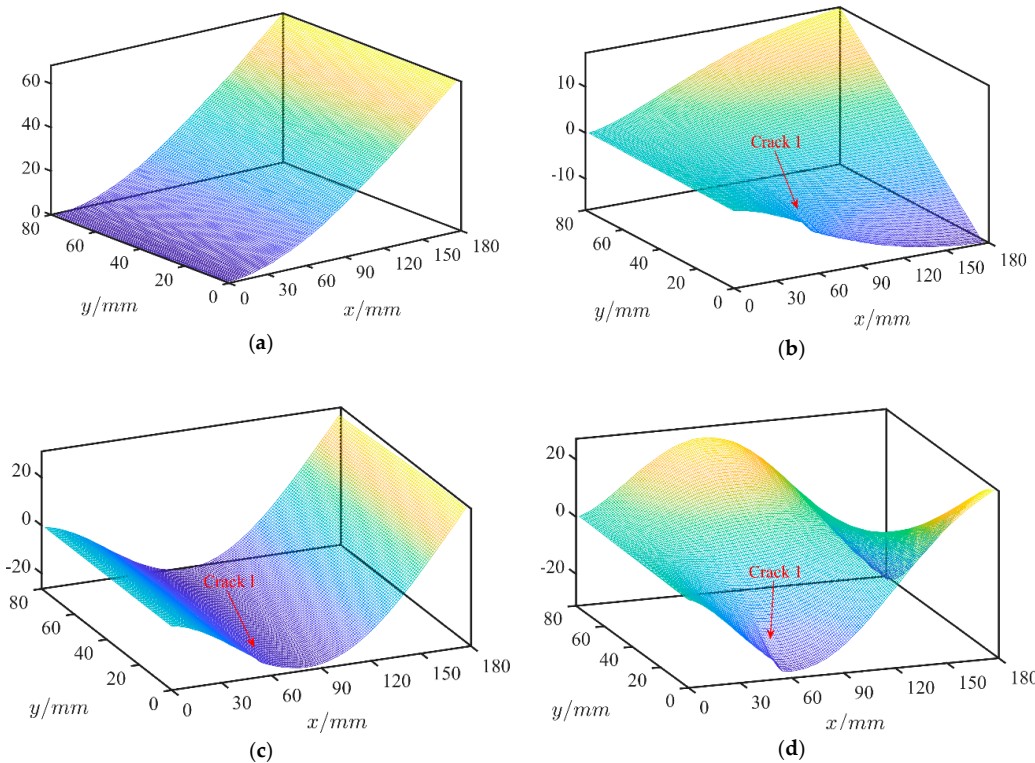

**Figure 7.** The first four mode shapes of a single crack plate using uniform-rate scanning CSLDV. (**a**) The first bending mode at 49.02 Hz. (**b**) The first torsion mode at 237.70 Hz. (**c**) The second bending mode at 311.52 Hz. (**d**) The second torsion mode at 781.25 Hz.

Without a priori knowledge of the undamaged plate, the undamaged mode shapes were extracted from the damaged ones using the polynomial function fitting method. The fit values of these modes against the polynomial orders are plotted in Figure 8. It can be seen that the fit value increased rapidly at lower orders and gradually converged at higher orders. Based on the convergence threshold of fit value, the minimum number of polynomial coefficients of the undamaged mode shapes were determined as (x = 9, y = 3) at 49.02 Hz, (x = 10, y = 10) at 237.70 Hz, (x = 9, y = 7) at 311.52 Hz, and (x = 13, y = 8) at 781.25 Hz, respectively.

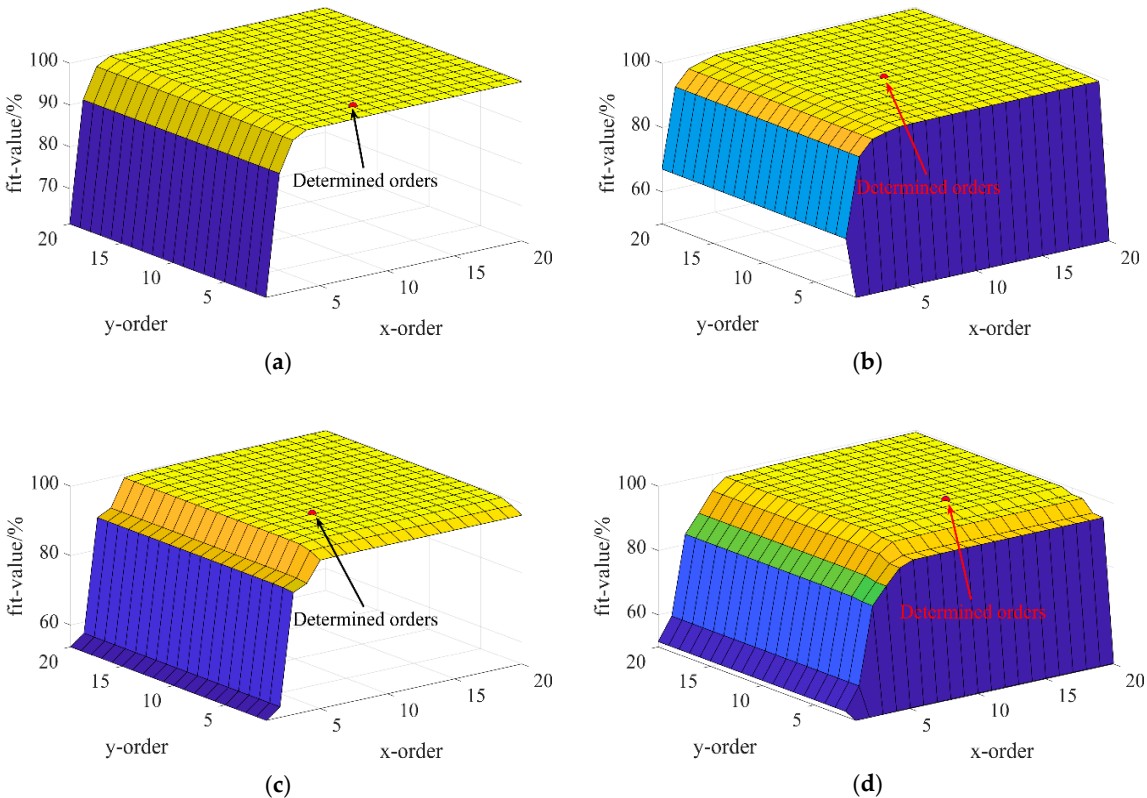

**Figure 8.** The fit value of first four modes against polynomial function orders. (**a**) The first bending mode at 49.02 Hz. (**b**) The first torsion mode at 237.70 Hz. (**c**) The second bending mode at 311.52 Hz. (**d**) The second torsion mode at 781.25 Hz.

Based on the damaged and the reconstructed undamaged mode shapes, the modal rotational mode shapes of the x and y directions with the interval spacing 1.0 mm were derived and the corresponding normalized modal rotational damage indicators (RDI) of these modes were plotted in Figure 9. It is shown that the modal rotational damage indicators of the first mode are very messy in both the x and y directions and cannot identify any crack location. The reason is attributed to the fact that the first mode shape is not sensitive to the crack and the tiny changes of displacement mode shape can be easily interfered with by the noise. But for other three mode shapes, the Crack 1 location can be located by the modal rotational damage indicators in both directions. Moreover, the RDI of y direction is much clearer and more accurate. Not only the crack location, but also the crack span can be clearly identified. This is consistent with the conclusion of simulation analysis.

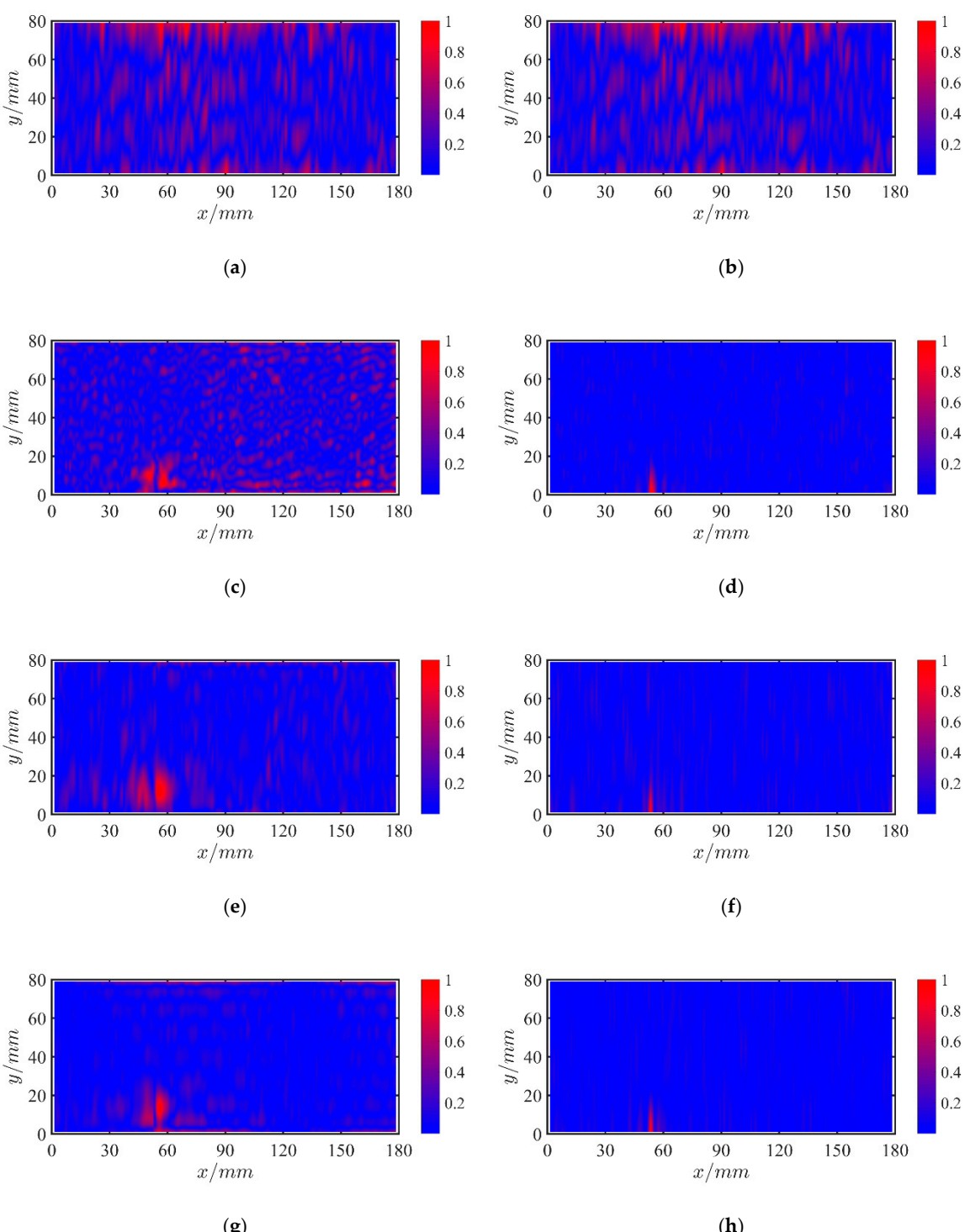

**Figure 9.** The RDIs colored plot of first four modes of a single crack plate. (**a**) RDIs of the first mode in the x direction. (**b**) RDIs of the first mode in the y direction. (**c**) RDIs of the second mode in the x direction. (**d**) RDIs of the second mode in the y direction. (**e**) RDIs of the third mode in the x direction. (**f**) RDIs of the third mode in the y direction. (**g**) RDIs of the fourth mode in the x direction. (**h**) RDIs of the fourth mode in the y direction.

### 4.2.2. Case 2: Multi Cracks

For Case 2, the corresponding first four mode frequencies are 49.61 Hz, 234.38 Hz, 299.61 Hz, and 717.58 Hz. The displacement mode shapes at these frequencies are plotted in Figure 10, respectively.

Results are the same as Case 1 and only the torsional modes can clearly reveal the discontinuity of the displacement mode shapes at the crack locations. Using the same procedure, mode shapes of the undamaged state can be reconstructed with determined function orders based on the fit value convergence threshold.

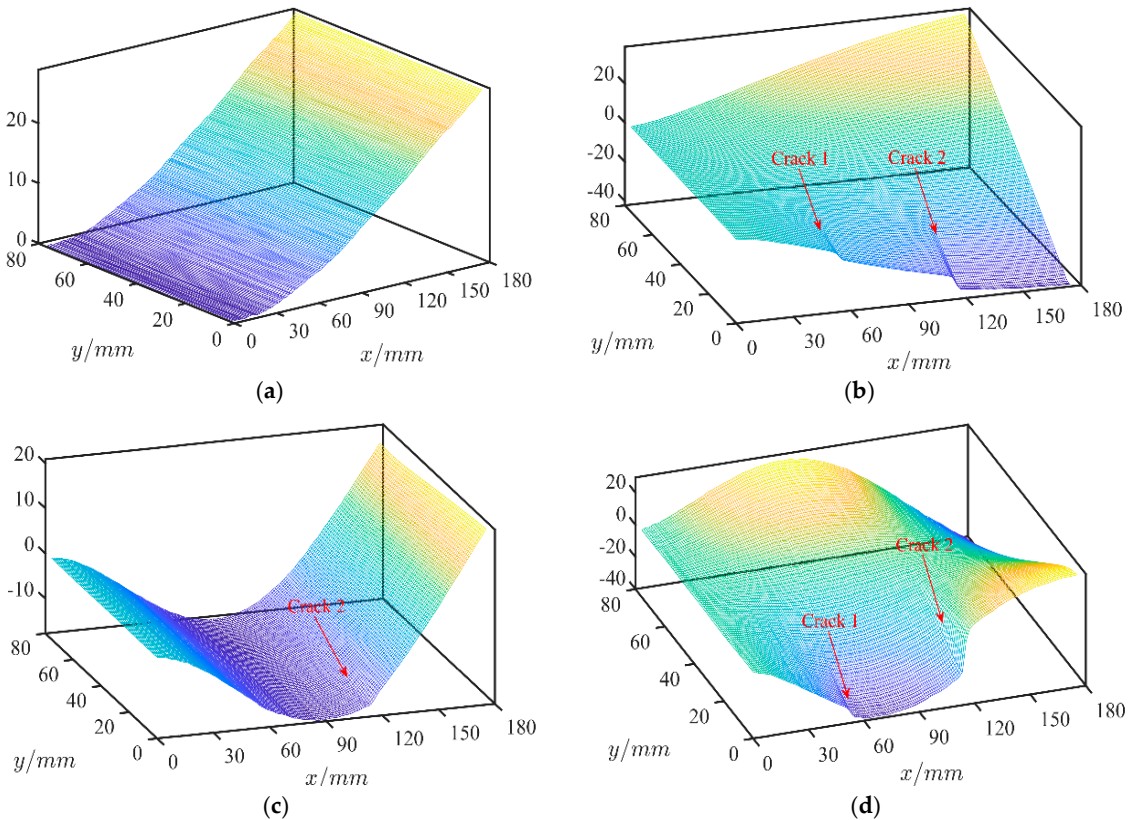

**Figure 10.** The first four measured mode shapes of the multi cracks plate using uniform-rate scanning CSLDV. (**a**) The first bending mode. (**b**) The first torsion mode. (**c**) The second bending mode. (**d**) The second torsion mode.

The normalized modal rotational damage indicators are plotted in Figure 11. For the first mode, the RDIs failed to detect and localize the damage as this mode is the bending mode and the modal rotational mode shapes in the x direction are close to zero. Therefore, RDIs of this mode are actually noises. For the third mode, the RDIs can only reflect the damage caused by Crack 2. This is because Crack 2 is much larger than Crack 1 and located at the large amplitude of vibration mode. However, modal rotational damage indicators of the torsional modes in the y direction are much better to identify the damage locations clearly, except the RDIs of the torsional modes in the x direction. This is because changes of the modal rotational mode shapes are mainly along the y direction and therefore the modal rotational mode shapes in the x direction can be easily contaminated by noise. From the RDIs of the second mode in the y direction, two cracks can be clearly located and the indicators at Crack 2 position are more significant than those at Crack 1 position, as shown in Figure 11d. It indicates that the damage severity can also be identified. However, the results derived from the fourth mode seem hardly to identify the location of Crack 1. It is because the damage indicators are normalized by the maximum value and illustrated by color plots, so the color of the relatively small Crack 1 is much lighter than that of the major crack. If the area is further zoomed, the surface plot of the rotational damage indicator of the second torsional mode in the y direction is replotted in Figure 12. It clearly shows that the damage indicator of Crack 1 is less than 20% of the indicator of Crack 2. When the left bottom quarter of the figure is enlarged separately, the indicator of Crack 1 is quite obvious. Therefore, the local scanning and damage detection can effectively identify the relatively small crack of the multiple damage case.

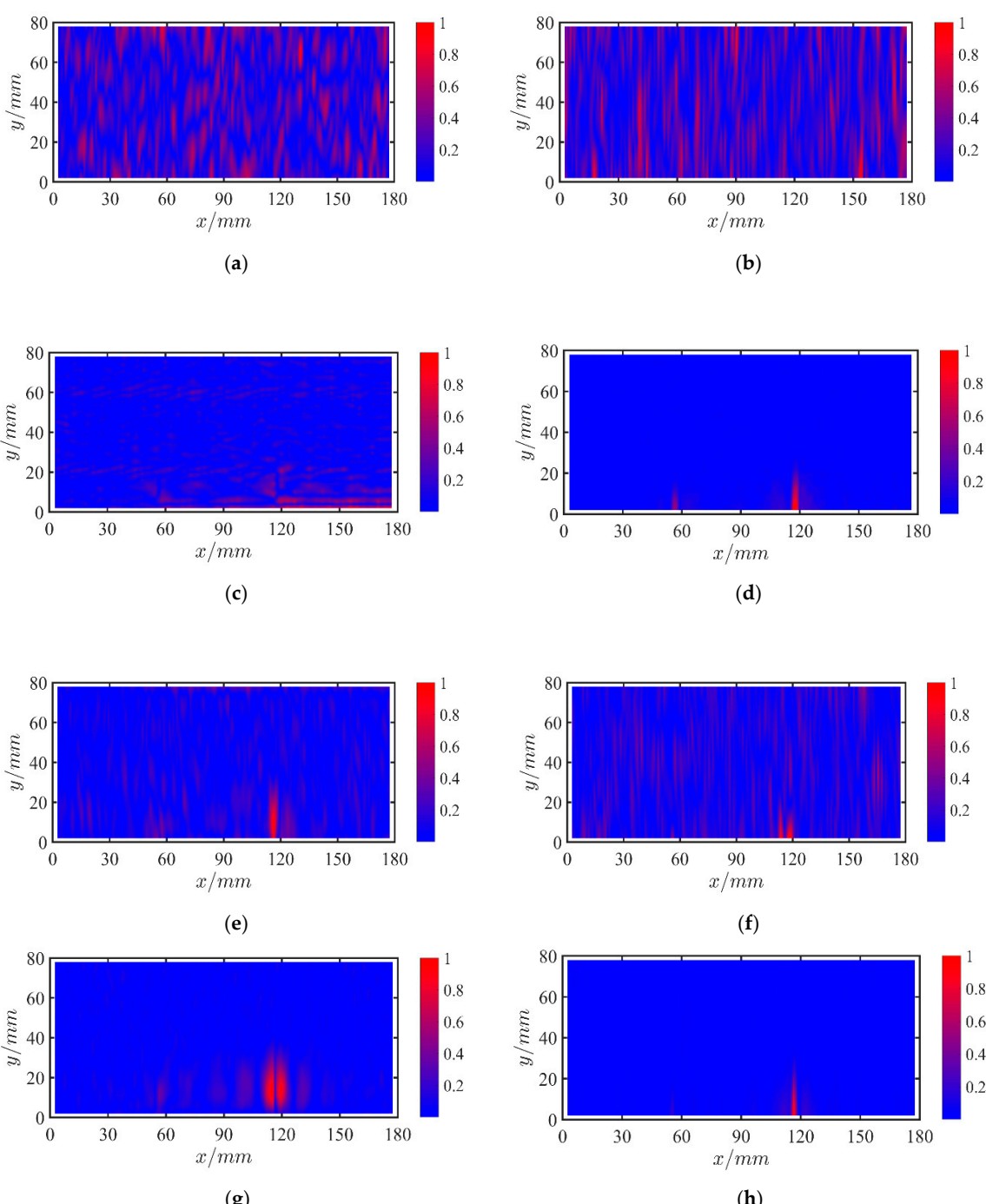

**Figure 11.** The RDIs colored plot of multi crack plate in the x and y directions. (**a**) RDIs of the first mode in the x direction. (**b**) RDIs of the first mode in the y direction. (**c**) RDIs of the second mode in the x direction. (**d**) RDIs of the second mode in the y direction. (**e**) RDIs of the third mode in the x direction. (**f**) RDIs of the third mode in the y direction. (**g**) RDIs of the fourth mode in the x direction. (**h**) RDIs of the fourth mode in the y direction.

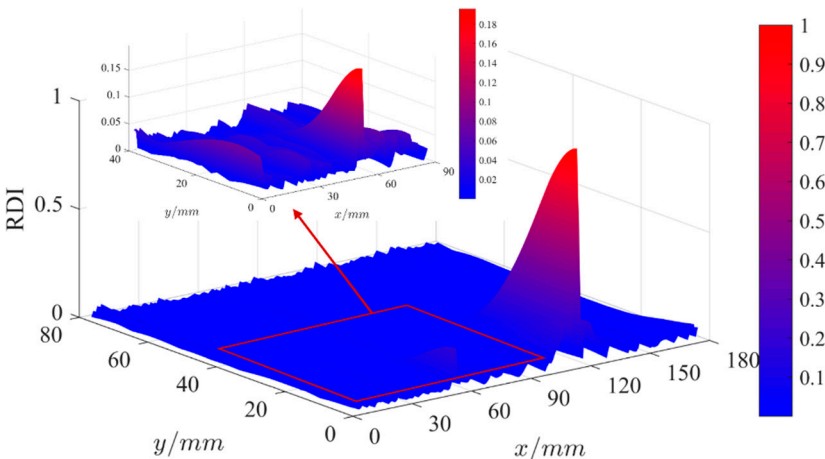

**Figure 12.** The RDIs surface plot of the second torsional mode in the y direction for the multi crack plate.

### 4.2.3. Case 3: Slight Crack

Investigations showed that only the torsional modes were sensitive to the crack damage. For a slight crack with 6 mm length, the measurement was undertaken with the scan rates of 0.02 Hz in the x direction and 4.0 Hz in the y direction at the frequencies of the first two torsional modes that are 233.43 Hz and 750.94 Hz, respectively. The same procedure was applied to obtain the two mode shapes and corresponding rotational damage indicators. The corresponding rotational damage indicators in the y direction are plotted in Figure 13a,b, respectively. Both figures clearly and correctly reveal the location and span direction of the slight crack.

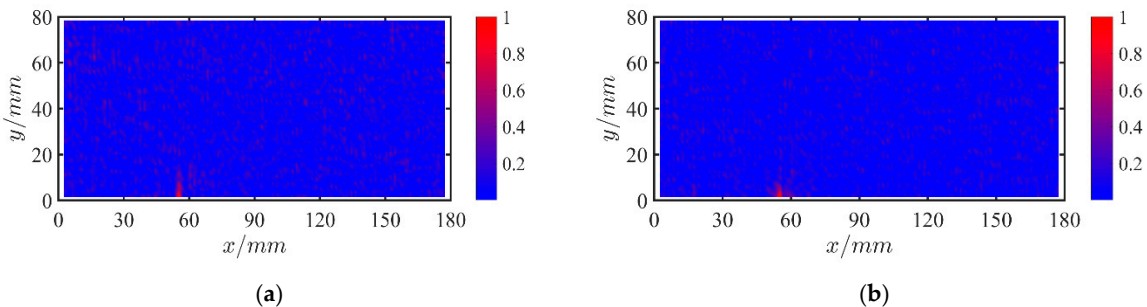

(**a**) (**b**)

**Figure 13.** The RDIs colored plot of the torsion modes in the y direction for a slight crack plate. (**a**) RDIs of the first torsion mode of 233.43 Hz. (**b**) RDIs of the second torsion mode of 750.94 Hz.

Actually, the scan frequency rate has a significant influence on measurement of the local slight change of the mode shape. The higher scan rate means the laser spot across the damage area very quickly and may not be able to capture the tiny local anomaly caused by damage. Moreover, the spatial resolution of measured shape data with a higher scan frequency rate also may not be able to reflect any change of the mode shape. In order to analyze the impact of the scan frequency rate, the scan frequency in the x direction was increased from 0.02 Hz to 0.05 Hz and the y direction increased from 4.0 Hz to 10.0 Hz to perform the measurement. The rotational damage indicators of the two torsional modes in the y direction are plotted in Figure 14. From the first plot, the crack position can be roughly located but the noise has some interference to the results. Compared with the results of 0.02 Hz scan rate, the impact of scan rates of the second torsional mode is much more serious, as shown in Figure 14b. According to these figures, the crack cannot be identified clearly. The results show that the lower scan rates are preferred for slight crack damage detection.

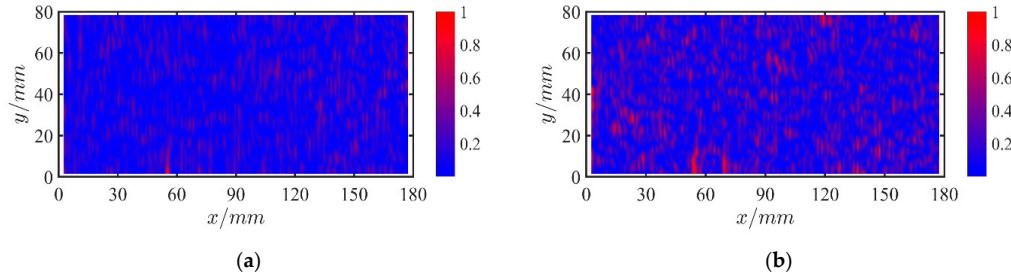

**Figure 14.** The RDI colored plot of the slight crack plate with the scanning rates of 0.05 Hz at the x direction and 10 Hz at the y direction. (**a**) The first torsion mode of 233.43 Hz. (**b**) The second torsion mode of 750.94 Hz.

## 5. Conclusions

In this paper, the damage detection method using the modal rotational mode shapes obtained from a uniform rate CSLDV technique is presented. The proposed Zigzag continuously scanning strategy is capable to acquire the mode shape data of the tested structure with any spatial resolution based on the mode shape demodulation method. This approach plays a paramount role in further damage detection. Without a priori knowledge of the undamaged structure, a polynomial function fitting method is proposed to reconstruct the ODS of undamaged state in order to meet the requirement to establish a damage indicator. In addition, a modal rotational damage indicator (RDI) is proposed and is used to detect and localize the cracks of a clamped plate structure. Investigations of a numerical simulation and experimental tests validated the proposed method. Results show that three cases with different cracks on the plate can be correctly identified using the experimental measurement data.

The modal rotational mode shapes are much more sensitive to the structural damage detection than the displacement mode shapes because they are the first derivative of the detailed displacement mode shapes of transitional DOFs with respect to the orthogonal directions. However, its accuracy relies on the spatial resolution and noise contamination in the measurement. With the development of laser technique, the uniform rate CSLDV method can greatly improve the accuracy and spatial resolution of the mode shape measurement. If the very low scan frequency is selected, the rotational damage indicator may clearly identify the local anomaly features caused by the slight damage. Therefore, damage detection based on the modal rotational mode shapes can be widely used in practice. The CSLDV techniques make damage detection effective and efficient, especially for ultra-light structures or structures with composite materials due to the features of the noncontact and full field measurement. In addition, any uncertainties in the measurement may affect the damage detection. For example, high temperatures will produce heat radiation on the surface of the structure and therefore introduce measurement noise when using CSLDV. Thus, the effect of noise on the damage detection must be eliminated before the damage detection technique is applied. Such work is underway and the results will be reported in the future. In another way, the change of boundary conditions will affect the dynamics of the structure but have little effect on the damage detection indicator since the proposed method is only based on the measurement of the damaged structure. The mode shape of the undamaged structure is obtained by polynomial function fitting using the measured mode shape data of the damaged state. This is one advantage of the proposed method.

**Author Contributions:** Conceptualization and Methodology, Z.H. and C.Z.; Numerical Analysis and Experimental validation, Z.H.; writing—original draft preparation, Z.H.; writing—review and editing, C.Z.

**Funding:** This research was funded by the National Natural Science Foundation of China and National Safety Academic Foundation of China, grant number U1730129.

**Acknowledgments:** The authors gratefully appreciate the supports from Jiangsu Province Key Laboratory of Aerospace Power System, the Key Laboratory of Aero-engine Thermal Environment and Structure, Ministry of Industry and Information Technology are also gratefully acknowledged.

**Conflicts of Interest:** The authors declare no conflict of interest.

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
