# Peer review of "Damage Detection Using Modal Rotational Mode Shapes Obtained with a Uniform Rate CSLDV Measurement"

_applsci, doi:10.3390/app9234982_

Round 1

Reviewer 1 Report

The present paper is aimed at studying an innovative method for damage detction by a

 continuously scanning laser Doppler vibrometer. The subject of the paper is of good interest, and the paper is generally well written. The methods are clearly presented, and results rationally discussed. Therefore, I can suggest publication of the paper, after some minor revisions.

Carefully check the paper for typo and grammar mistakes. Throughout the paper, a lot of sentences are not correctly written In the conclusion, it is not clear why the developed method is particularly suitable for ultra-light structures. In addition, the authors should elaborate more in detail the suitability to composite structures. Is it likely that anisotropy can play a crucial role in the developed measurement method?

Reviewer 2 Report

The paper presents a vibration based damage detection of plate-like structures using CSLDV for measurements. The paper provides validation on numerical model as well as experimental validation for a steel plate with a crack.

The paper has a potential to have a decent impact and be of interest to the research community but significant work needs to be carried out before the paper can reach a level suitable for publication.

The major comments on the paper are:

The literature review has covered several relevant publications but there are few areas which are relevant and have been missed.

For instance the long gauge strain sensors are known to measure the rotational mode shape of the beam and have been used by several researchers for improved damage detection. Some of this work has to be cited.

There has been some work by Sazunov et al for the optimal sampling for computation of mode shape derivatives.

Also as the authors are using only the first 4 modes of vibration for damage, the popular assumption mostly in civil engineering applications is the use of flexibility indices for damage detection as they give good results with limited mode information. So a discussion highlighting this and why the authors used curvature may be interesting to the reader. Also for flexibility index a paper by Soman et al. may be relevant.

Soman, R., Kyriakides, M., Onoufriou, T., & Ostachowicz, W. (2018). Numerical evaluation of multi-metric data fusion based structural health monitoring of long span bridge structures. Structure and Infrastructure Engineering14(6), 673-684.

The methodology needs to be explained in detail. The reviewer had to read the methodology section several times in order to understand the method clearly. Considerable work needs to be done on this section to improve the understanding and the readability

The presentation of results for the numerical and experimental needs to be improved as well. The figure (3 and 7) are not very clear, may be applying a grid and using a different angle for the results may be useful.

Their is no mention of any uncertainties in the measurement including temperature, and the changes introduced in the boundary conditions when the crack was introduced. The discussion on the shortcoming of the methodology in this area is necessary.

Also how was it ensured that the grid of points for healthy and damaged condition were indeed the same. Especially in the regions around the nodes of the modes even a minor error can lead to wrong results. A discussion on this is necessary.

apart from this there are several grammatical mistakes, typos and errors in spelling. A thorough proofreading is necessary.

I expect the authors to provide point by point response to the major concerns in the paper

Round 2

Reviewer 2 Report

Most of the changes introduced in the paper are satisfactory.

In the eyes of the reviewer there are still some issues that need to be addressed.

The language needs to be improved significantly. There are several grammatical errors. A thorough proof reading or using the editing services is recommended.

The claim made in 450 needs to be supported with appropriate reference. This is the first time the reviewer has come across the mention of this phenomena. The point made about thermal effects was more due to change in the properties of the material, expansion etc.

Also the reason why this method is identified for light weight structures is not yet obvious.

The highlights of the paper should be explicitly mentioned.

Author Response

Most of the changes introduced in the paper are satisfactory. In the eyes of the reviewer there are still some issues that need to be addressed.

The language needs to be improved significantly. There are several grammatical errors. A thorough proof reading or using the editing services is recommended.

Response: We thoroughly read through our manuscript and made corrections about grammatical errors.

The claim made in 450 needs to be supported with appropriate reference. This is the first time the reviewer has come across the mention of this phenomena. The point made about thermal effects was more due to change in the properties of the material, expansion etc. Also the reason why this method is identified for light weight structures is not yet obvious.

Response: Thank you for clarifying this point. We suppose the misunderstanding is due to our simplified explanation in the revision. Now we have reworded the sentences and add more detailed explanation. See the revised manuscript.

The highlights of the paper should be explicitly mentioned.

Response: The highlights of the paper are addressed in abstract, conclusion and also in the introduction part.